# Multi-agent Deep FBSDE Representation For Large Scale Stochastic Differential Games

## Abstract

In this paper we present a deep learning framework for solving large-scale multi-agent non-cooperative stochastic games using fictitious play. The Hamilton-Jacobi-Bellman (HJB) PDE associated with each agent is reformulated into a set of Forward-Backward Stochastic Differential Equations (FBSDEs) and solved via forward sampling on a suitably defined neural network architecture. Decision making in multi-agent systems suffers from curse of dimensionality and strategy degeneration as the number of agents and time horizon increase. We propose a novel Deep FBSDE controller framework which is shown to outperform the current state-of-the-art deep fictitious play algorithm on a high dimensional interbank lending/borrowing problem. More importantly, our approach mitigates the curse of many agents and reduces computational and memory complexity, allowing us to scale up to 1,000 agents in simulation, a scale which, to the best of our knowledge, represents a new state of the art. Finally, we showcase the framework's applicability in robotics on a belief-space autonomous racing problem.

## 1 Introduction

Stochastic differential games represent a framework for investigating scenarios where multiple players make decisions while operating in a dynamic and stochastic environment. The theory of differential games dates back to the seminal work of Isaacs (1965) studying two-player zero-sum dynamic games, with a first stochastic extension appearing in Kushner & Chamberlain (1969). A key step in the study of games is obtaining the Nash equilibrium among players (Osborne & Rubinstein, 1994). A Nash equilibrium represents the solution of non-cooperative game where two or more players are involved. Each player cannot gain benefit by modifying his/her own strategy given opponents equilibrium strategy. In the context of adversarial multi-objective games, the Nash equilibrium can be represented as a system of coupled Hamilton-Jacobi-Bellman (HJB) equations when the system satisfies the Markovian property. Analytic solutions exist only for few special cases. Therefore, obtaining the Nash equilibrium solution is usually done numerically, and this can become challenging as the number of states/agents increases. Despite extensive theoretical work, the algorithmic part has received less attention and mainly addresses special cases of differential games (e.g., Duncan & Pasik-Duncan (2015)), or suffers from the curse of dimensionality (Kushner, 2002). Nevertheless, stochastic differential games have a variety of applications including in robotics and autonomy, economics and management. Relevant examples include Mataramvura & Øksendal (2008), which formulate portfolio management as a stochastic differential game in order to obtain a market portfolio that minimizes the convex risk measure of a terminal wealth index value, as well as Prasad & Sethi (2004), who investigate optimal advertising spending in duopolistic settings via stochastic differential games.

Reinforcement Learning (RL) aims in obtaining a policy which can generate optimal sequential decisions while interacting with the environment. Commonly, the policy is trained by collecting histories of states, actions, and rewards, and updating the policy accordingly. Multi-agent Reinforcement Learning (MARL) is an extension of RL where several agents compete in a common environment, which is a more complex task due to the interaction between several agents and the environment, as well as between the agents. One approach is to assume agents to be part of environment (Tan, 1993), but this may lead to unstable learning during policy updates (Matignon et al., 2012). On the other hand, a centralized approach considers MARL through an augmented state and action system, reducing its training to that of single agent RL problem. Because of the combinatorial complexity,

the centralized learning method cannot scale to more than 10 agents (Yang et al., 2019). Another method is centralized training and decentralized execute (CTDE), however the challenge therein lies on how to decompose value function in the execute phase for value-based MARL. Sunehag et al. (2018) and Zhou et al. (2019) decompose the joint value function into a summation of individual value functions. Rashid et al. (2018) keep the monotonic trends between centralized and decentralized value functions by augmenting the summation non-linearly and designing a mixing network (QMIX). Further modifications on QMIX include Son et al. (2019); Mahajan et al. (2019).

The mathematical formulation of a differential game leads to a nonlinear PDE. This motivates algorithmic development for differential games that combine elements of PDE theory with deep learning. Recent encouraging results (Han et al., 2018; Raissi, 2018) in solving nonlinear PDEs within the deep learning community illustrate the scalability and numerical efficiency of neural networks. The transition from a PDE formulation to a trainable neural network is done via the concept of a system of Forward-Backward Stochastic Differential Equations (FBSDEs). Specifically, certain PDE solutions are linked to solutions of FBSDEs, and the latter can be solved using a suitably defined neural network architecture. This is known in the literature as the *deep FBSDE* approach. Han et al. (2018); Pereira et al. (2019); Wang et al. (2019b) utilize various deep neural network architectures to solve such stochastic systems. However, these algorithms address single agent dynamical systems. Two-player zero-sum games using FBSDEs were initially developed in Exarchos et al. (2019) and transferred to a deep learning setting in Wang et al. (2019a). Recently,Hu (2019) brought deep learning into fictitious play to solve multi-agent non-zero-sum game, Han & Hu (2019) introduced the deep FBSDEs to a multi-agent scenario and the concept of fictitious play, furthermore, Han et al. (2020) gives the convergence proof.

In this work we propose an alternative deep FBSDE approach to multi-agent non-cooperative differential games, aiming on reducing complexity and increasing the number of agents the framework can handle. The main contribution of our work is threefold:

1. We introduce an efficient Deep FBSDE framework for solving stochastic multi-agent games via fictitious play that outperforms the current state of the art in Relative Square Error (RSE) and runtime/memory efficiency on an inter-bank lending/borrowing example.

2. We demonstrate that our approach scales to a much larger number of agents (up to 1,000 agents, compared to 50 in existing work). To the best of our knowledge, this represents a new state of the art.

3. We showcase the applicability of our framework to robotics on a belief-space autonomous racing problem which has larger individual control and state space. The experiments demonstrates that the decoupled BSDE provides the possibility of applications for competitive scenario.

The rest of the paper is organized as follows: in Section 2 we present the mathematical preliminaries. In Section 3 we introduce the Deep Fictitious Play Belief FBSDE, with simulation results following in Section 4. We conclude the paper and discuss some future directions in Section 5.

## 2 MULTI-AGENT FICTITIOUS PLAY FBSDE

Fictitious play is a learning rule first introduced in Brown (1951) where each player presumes other players' strategies to be fixed. An $N$-player game can then be decoupled into $N$ individual decision-making problems which can be solved iteratively over $M$ stages. When each agent[1] converges to a stationary strategy at stage $m$, this strategy will become the stationary strategy for other players at stage $m + 1$. We consider a $N$-player non-cooperative stochastic differential game with dynamics

$$\mathrm{d}\boldsymbol{X}(t) = \big(f(\boldsymbol{X}(t), t) + G(\boldsymbol{X}(t), t)\boldsymbol{U}(t)\big)\mathrm{d}t + \Sigma(\boldsymbol{X}(t), t)\mathrm{d}\boldsymbol{W}(t), \quad \boldsymbol{X}(0) = \boldsymbol{X}_0, \qquad (1)$$

where $\boldsymbol{X} = (\boldsymbol{x}_1, \boldsymbol{x}_2, \ldots, \boldsymbol{x}_N)$ is a vector containing the state process of all agents generated by their controls $\boldsymbol{U} = (\boldsymbol{u}_1, \boldsymbol{u}_2, \ldots, \boldsymbol{u}_N)$ with $\boldsymbol{x}_i \in \mathbb{R}^{n_x}$ and $\boldsymbol{u}_i \in \mathbb{R}^{n_u}$. Here, $f : \mathbb{R}^{n_x} \times [0, T] \to \mathbb{R}^{n_x}$ represents the drift dynamics, $G : \mathbb{R}^{n_x} \times [0, T] \to \mathbb{R}^{n_x \times n_u}$ represents the actuator dynamics, and $\Sigma : [0, T] \times \mathbb{R}^n \to \mathbb{R}^{n_x \times n_w}$ represents the diffusion term. We assume that each agent is only driven by its own controls so $G$ is a block diagonal matrix with $G_i$ corresponding to the actuation of agent $i$.

---

[1]Agent and player are used interchangeably in this paper

Each agent is also driven by its own $n_w$-dimensional independent Brownian motion $W_i$, and denote $\boldsymbol{W} = (W_1, W_2, \ldots, W_N)$.

Let $\mathbb{U}_i$ be the set of admissible strategies for agent $i \in \mathbb{I} := \{1, 2, \ldots, N\}$ and $\mathbb{U} = \otimes_{i=1}^{N} \mathbb{U}_i$ as the Kronecker product space of $\mathbb{U}_i$. Given the other agents' strategies, the stochastic optimal control problem for agent $i$ under the fictitious play assumption is defined as minimizing the expectation of the cumulative cost functional $J_t^i$

$$J_t^i(\boldsymbol{X}, \boldsymbol{u}_{i,m}; \boldsymbol{u}_{-i,m-1}) = \mathbb{E}\left[g(\boldsymbol{X}(T)) + \int_t^T C^i(\boldsymbol{X}(\tau), \boldsymbol{u}_{i,m}(\boldsymbol{X}(\tau), \tau), \tau; \boldsymbol{u}_{-i,m-1})\mathrm{d}\tau\right], \quad (2)$$

where $g : \mathbb{R}^{n_x} \to \mathbb{R}^+$ is the terminal cost, and $C^i : [0, T] \times \mathbb{R}^{n_x} \times \mathbb{U} \to \mathbb{R}^+$ is the running cost for the $i$-th player. In this paper we assume that the running cost is of the form $C(\boldsymbol{X}, \boldsymbol{u}_{i,m}, t) = q(\boldsymbol{X}) + \frac{1}{2}\boldsymbol{u}_{i,m}^{\mathrm{T}} R \boldsymbol{u}_{i,m} + \boldsymbol{X}^{\mathrm{T}} Q \boldsymbol{u}_{i,m}$. We use the double subscript $\boldsymbol{u}_{i,m}$ to denote the control of agent $i$ at stage m and the negative subscript $-i$ as the strategies excluding player $i$, $\boldsymbol{u}_{-i} = (\boldsymbol{u}_1, \ldots, \boldsymbol{u}_{i-1}, \boldsymbol{u}_{i+1}, \ldots, \boldsymbol{u}_N)$. We can define value function of each player as

$$V^i(t, \boldsymbol{X}(t)) = \inf_{\boldsymbol{u}_{i,m} \in \mathbb{U}_i} \left[J_t^i(\boldsymbol{X}, \boldsymbol{u}_{i,m}; \boldsymbol{u}_{-i,m-1})\right], \quad V^i(T, \boldsymbol{X}(T)) = g(\boldsymbol{X}(T)). \quad (3)$$

Assume that the value function in eq. (3) is once differentiable w.r.t. $t$ and twice differentiable w.r.t. $x$. Then, standard stochastic optimal control theory leads to the HJB PDE

$$V^i + h + V_x^{i\mathrm{T}}(f + G\boldsymbol{U}_{0,-i}) + \frac{1}{2}\mathrm{tr}(V_{xx}^i \Sigma\Sigma^{\mathrm{T}}) = 0, \quad V^i(T, \boldsymbol{X}) = g(\boldsymbol{X}(T)), \quad (4)$$

where $h = C^{i*} + G\boldsymbol{U}_{*,0}$. The double subscript of $U_{*,0}$ denotes the augmentation of the optimal control $\boldsymbol{u}_{i,m}^* = -R^{-1}(G_i^{\mathrm{T}} V_x^i + Q_i^{\mathrm{T}} x)$ and zero control $\boldsymbol{u}_{-i,m-1} = 0$, and $\boldsymbol{U}_{0,-i}$ denotes the augmentation of $\boldsymbol{u}_{i,m} = 0$ and $\boldsymbol{u}_{-i,m-1}$. Here we drop the functional dependencies in the HJB equation for simplicity. The detailed proof is in Appendix A. The value function in the HJB PDE can be related to a set of FBSDEs

$$\begin{aligned}
\mathrm{d}\boldsymbol{X} &= (f + G\boldsymbol{U}_{*,-i})\mathrm{d}t + \Sigma\mathrm{d}\boldsymbol{W}, \quad \boldsymbol{X}(0) = \boldsymbol{x}_0 \\
\mathrm{d}V^i &= -(h + V_x^{i\mathrm{T}} G\boldsymbol{U}_{*,0})\mathrm{d}t + V_x^{\mathrm{T}}\Sigma\mathrm{d}W, \quad V(T) = g(\boldsymbol{X}(T)),
\end{aligned} \quad (5)$$

where the backward process corresponds to the value function. The detailed derivation can be found in Appendix B. Note that the FBSDEs here differ from that of Han & Hu (2019) in the optimal control of agent $i$, $G\mathbb{U}_{*,-i}$, in the forward process and compensation, $V_x^{i\mathrm{T}} G\boldsymbol{U}_{*,0}$, in the backward process. This is known as the importance sampling for FBSDEs and allows for the FBSDEs to be guided to explore the state space more efficiently.

## 3 DEEP FICTITIOUS PLAY FBSDE CONTROLLER

In this section, we introduce a novel and scalable Deep Fictitious Play FBSDE (SDFP) Controller to solve the multi-agent stochastic optimal control problem. The framework can be extended to the partially observable scenario by combining with an Extended Kalman Filter, whose belief propagation can be described by an SDE for the mean and variance (see derivation in Appendix C). By the natural of decoupled BSDE, the framework can also been extended to cooperative and competitive scenario. In this paper, we demonstrate the example of competitive scenario.

### 3.1 NETWORK ARCHITECTURE AND ALGORITHM

Inspired by the success of LSTM-based deep FBSDE controllers (Wang et al., 2019b; Pereira et al., 2019), we propose an approach based on an LSTM architecture similar to Pereira et al. (2019). The benefits of introducing LSTM are two-fold: 1) LSTM can capture the features of sequential data. A performance comparison between LSTM and fully connected (FC) layers in the deep FBSDE framework has been elaborated in Wang et al. (2019b); 2) LSTM significantly reduces the memory complexity of our model since the memory complexity of LSTM with respect to time is $\mathcal{O}(1)$ in the inference phase compared with $\mathcal{O}(T)$ in previous work (Han et al., 2018), where $T$ is the number of time steps. The overall architecture of SDFP is shown in Fig. 1 and features the same time discretization scheme as Pereira et al. (2019). Each player's policy is characterized by its own copy

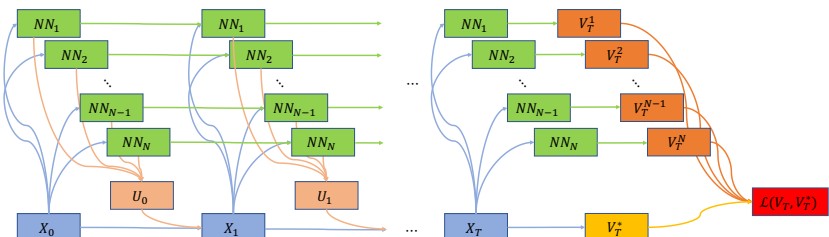

Figure 1: SDFP framework for $N$ Players. Each NN block has the architecture in Fig. 11.

of the network defined in Fig. 11. At stage $m$, each player can access the stationary strategy of all other players from stage $m-1$. During training within a stage, the initial value of each player is predicted by a FC layer parameterized by $\phi$. At each timestep, the optimal policy for each player is computed using the value function gradient prediction $V_x^i$ from the recurrent network (consisting of FC and LSTM layers), parameterized by $\theta$. The FSDE and BSDE are then forward-propagated using the Euler integration scheme. At terminal time $T$, the loss function for each player is constructed as the mean squared error between the propagated terminal value $V_T^i$ and the true terminal value $V_T^{i*}$ computed from the terminal state. The parameters $\phi$ and $\theta$ of each player can be trained using any stochastic gradient descent type optimizer such as Adam. The detailed training procedure is shown in Algorithm 2.

### 3.2 MITIGATING CURSE OF DIMENSIONALITY AND SAMPLE COMPLEXITY

Scalability and sample efficiency are two crucial criteria of reinforcement learning. In SDFP, as the number of agents increases, the number of neural network copies would increase correspondingly. Meanwhile, the size of each neural network should be enlarged to gain enough capacity to capture the representation of many agents, leading to the infamous curse of dimensionality; this limits the scalability of prior works. However, one can mitigate the curse of dimensionality in this case by taking advantage of the symmetric game setup. We summarize merits of symmetric game as following:

1. Since all agents have the same dynamics and cost function, only one copy of the network is needed. The strategy of other agents can be inferred by applying the same network.

2. Thanks to the symmetric property, we can applied invariant layer to extract invariant features to accelerate training and improve the performance with respect to the accumulate cost and RSE loss.

**Sharing one network:** It's important to note that querying other agents should not introduce additional gradient paths. This significantly reduces the memory complexity. When querying other agents' strategy, one can either iterate through each agent or feed all agents' states to the network in a batch. The latter approach reduces the time complexity by adopting the parallel nature of modern GPU but requires $\mathcal{O}(N^2)$ memory rather than $\mathcal{O}(N)$ for the first approach.

**Invariant Layers:** The memory complexity can be further reduced with an invariant layer embedding (Zaheer et al., 2017). The invariant layer utilizes a sum function along with the features in the same set to render the network invariant to permutation of agents. We apply the invariant layer on $\boldsymbol{X}_{-i}$ and concatenate the resulting features to the features extracted from $\boldsymbol{X}_i$. However, vanilla invariant layer embedding will not reduce the memory complexity. Thanks to the symmetric problem setup, one can apply a trick to reduce the invariant layer memory complexity form $\mathcal{O}(N^2)$ to $\mathcal{O}(N)$. A

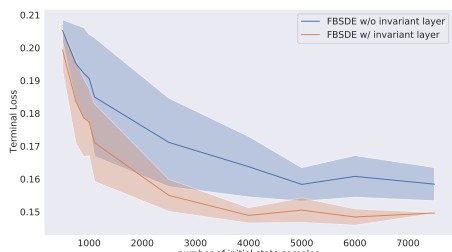

Figure 2: Sample efficiency between FBSDE framework w/ and w/o invariant layer.

detailed introduction to the invariant layer and our implementation can be found in Appendix D and E. The full algorithm is outlined in Algorithm 1.

---

**Algorithm 1** Scalable Deep Fictitious Play FBSDE for symmetric simplification

---

1: **Hyper-parameters**:$N$: Number of players, $T$: Number of timesteps, $M$: Number of stages in fictitious play, $N_{gd}$: Number of gradient descent steps per stage, $U_0$: the initial strategies for players in set $\mathbb{I}$, $B$: Batch size, $\epsilon$: training threshold, $\Delta t$: time discretization
2: **Parameters**:$V(\boldsymbol{x}_0; \phi)$: Network weights for initial value prediction, $\theta$: Weights and bias of fully connected layers and LSTM layers.
3: $\theta$: Initialize trainable papermeters:$\theta^0$, $\phi^0$
4: **while** LOSS is above certain threshold $\epsilon$ **do**
5:   **for** $m \leftarrow 1$ to $M$ **do**
6:     **for** all $i \in \mathbb{I}$ *in parallel* **do**
7:       Collect opponent agent's policy which is same as $i$th policy: $f^{m-1}_{LSTM_i}(\cdot), f^{m-1}_{FC_i}(\cdot)$
8:       **for** $l \leftarrow 1$ to $N_{gd}$ **do**
9:         **for** $t \leftarrow 1$ to $T-1$ **do**
10:           **if** Using Invariant Layer **then**
11:             $\boldsymbol{X}_t = f_{invariant}(\boldsymbol{X}_t)$
12:           **end if**
13:           **for** $j \leftarrow 1$ to $B$ *in parallel* **do**
14:             Compute network prediction for $i$th player: $V^j_{x_i,t} = f^m_{FC_i}(f^m_{LSTM_i}(\boldsymbol{X}^j_t; \theta^{l-1}_i))$
15:             Compute $i$th optimal Control :$\boldsymbol{u}^{j,\star}_{i,t} = -R^{-1}_i(G^{\mathrm{T}}_i V^j_{x_i,t} + Q^{\mathrm{T}}_i \boldsymbol{x}^j_i)$
16:             Infer $-i$th players' network prediction and stop the gradient for them: $V_{x_{-i},t} = f^{m-1}_{FC_i}(f^{m-1}_{LSTM_i}(\boldsymbol{X}_t; \theta_i))$
17:             Compute $-i$th optimal Control and stop the gradient for them: $\boldsymbol{u}^{j,\star}_{-i,t} = -R^{-1}_{-i}(G^{\mathrm{T}}_{-i} V^j_{x_{-i},t} + Q^{\mathrm{T}}_{-i} \boldsymbol{x}^j_{-i})$
18:             Sample noise $\Delta W^j \sim \mathcal{N}(0, \Delta t)$
19:             Propagate FSDE: $\boldsymbol{X}^j_{t+1} = f_{FSDE}(\boldsymbol{X}^j_t, \boldsymbol{u}^{j,\star}_{i,t}, \boldsymbol{u}^{j,*}_{-i,t}, \Delta W^j, t)$
20:             Propagate BSDE: $V^j_{i,t+1} = f_{BSDE}(\boldsymbol{X}^j_t, \boldsymbol{u}^{j,*}_{i,t}, \Delta W^j, t)$
21:           **end for**
22:         **end for**
23:         Compute loss: $\mathcal{L} = \frac{1}{B}\sum_{j=1}^B (V^{j,\star}_T - V^j_{i,T})^2$
24:         Gradient Update: $\theta^l, \phi^l$
25:       **end for**
26:     **end for**
27:   **end for**
28: **end while**

---

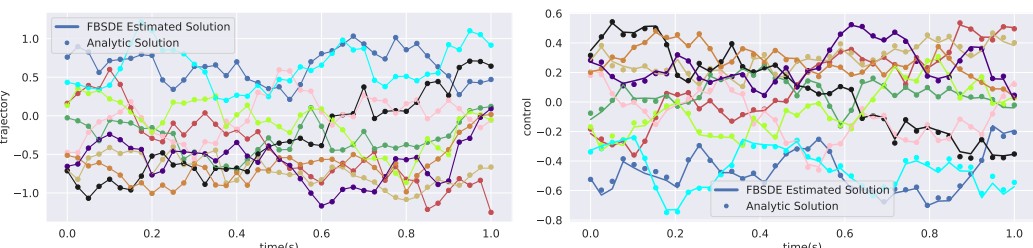

Figure 3: Comparison of SDFP and analytic solution for the inter-bank problem. Both the state (*left*) and control (*right*) trajectories are aligned with the analytic solution (represented by dots).

## 4   SIMULATION RESULTS

In this section, we demonstrate the capability of SFDP on two different systems in simulation. We first apply the framework to an inter-bank lending/borrowing problem, which is a classical multi-player non-cooperative game with an analytic solution. We compare against both the analytic solution and prior work (Han & Hu, 2019). Different approaches introduced in Section 3.2 are compared empirically on this system. We also apply the framework to a variation of the problem for which no analytic solution exists. Finally, we showcase the general applicability of our framework in an autonomous racing problem in belief space. All experiment configurations can be found in Ap-

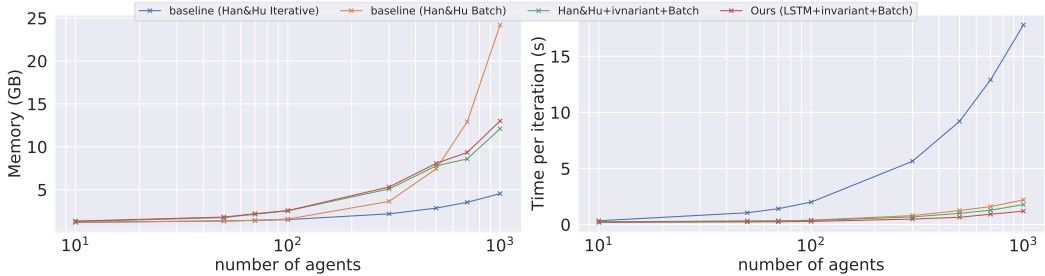

Figure 4: Time and memory complexity comparison between batch, iterate and invariant layer+batch implementations. Time complexity is measured by per-iteration time.

Table 1: Comparison with previous work on the 10-agent inter-bank problem.

| Framework | # stages | Learning Rate | RSE | Total Time (hr) [1] |
|---|---|---|---|---|
| Han & Hu (2019) | 80 | 1e-3 | 0.0423 | 2.23 |
| SFDP | 80 | 1e-3 | 0.0105 | 1.55 |

pendix J. we plot the results of 3 repeated runs with different seeds with the line and shaded region showing the mean and mean±standard deviation respectively. The hyperparameters and dynamics coefficients used in the inter-bank experiments are the same as Han & Hu (2019) unless otherwise noted.

### 4.1 INTER-BANK LENDING/BORROWING PROBLEM

We first consider an inter-bank lending and borrowing model (Carmona et al., 2013) where the dynamics of the log-monetary reserves of $N$ banks is described by the diffusion process

$$\mathrm{d}X_t^i = \left[a(\bar{X} - X_t^i) + u_t^i\right]\mathrm{d}t + \sigma(\rho \mathrm{d}W_t^0 + \sqrt{1-\rho^2}\mathrm{d}W_t^i), \bar{X}_t = \frac{1}{N}\sum_{i=1}^{N}X_t^i, i \in \mathbb{I}. \quad (6)$$

The state $X_t^i \in \mathbb{R}$ denotes the log-monetary reserve of bank $i$ at time $t > 0$. The control $u_t^i$ denotes the cash flow to/from a central bank, where as $a(\bar{X} - X_t^i)$ denotes the lending/borrowing rate of bank $i$ from all other banks. The system is driven by $N$ independent standard Brownian motion $W_t^i$, which denotes the idiosyncratic noise, and a common noise $W_t^0$. The cost function has the form,

$$C_{i,t}(\boldsymbol{X}, \boldsymbol{u}_i; \boldsymbol{u}_{-i}) = \frac{1}{2}\boldsymbol{u}_i^2 - q\boldsymbol{u}_i(\bar{\boldsymbol{X}} - \boldsymbol{X}_i) + \frac{\epsilon}{2}(\bar{\boldsymbol{X}} - \boldsymbol{X}_i)^2. \quad (7)$$

The derivation of the FBSDEs and analytic solution can be found in Appendix F. We compare the result of implementation corresponding to Algorithm 2 on a 10-agent problem with analytic solution and previous work from Han & Hu (2019) with the same hyperparameters. Fig. 3 shows the performance of our method compared with analytic solution. The state and control trajectories outputted by the deep FBSDE solution are aligned closely with the analytic solution. Table 1 shows the numerical performance compared with prior work by Han & Hu (2019). Our method outperforms by Relative Square Error (RSE) metrics and computation wall time. The RSE is defined as following:

$$RSE = \frac{\sum_{1 \le j \le B}^{i \in \mathbb{I}}(\hat{V}^i(0, \boldsymbol{X}^j(0)) - V^i(0, \boldsymbol{X}^j(0)))^2}{\sum_{1 \le j \le B}^{i \in \mathbb{I}}(\hat{V}^i(0, \boldsymbol{X}^j(0)) - \bar{V}^i(0, \boldsymbol{X}^j(0)))^2}, \quad (8)$$

Where $\hat{V}^i$ is the analytic solution of value function for $i$th agents at intial state $\boldsymbol{X}^j(0)$. The initial state $\boldsymbol{X}^j(0)$ is new batch of data sampled from same distribution as $\boldsymbol{X}(0)$ in the training phase. The batch size $B$ is 256 for all inter-bank simulations. $V^i$ is the approximated value function for $i$th agent by FBSDE controller, and $\bar{V}^i$ is the average of analytic solution for $i$th agent over the entire batch.

**Time/Memory Complexity Analysis:** We empirically verify the time and memory complexity of different implementation approaches introduced in 3.2, which is shown in Fig. 4. Note that all

---

[1]The experiment is conducted on Nvidia TITAN RTX

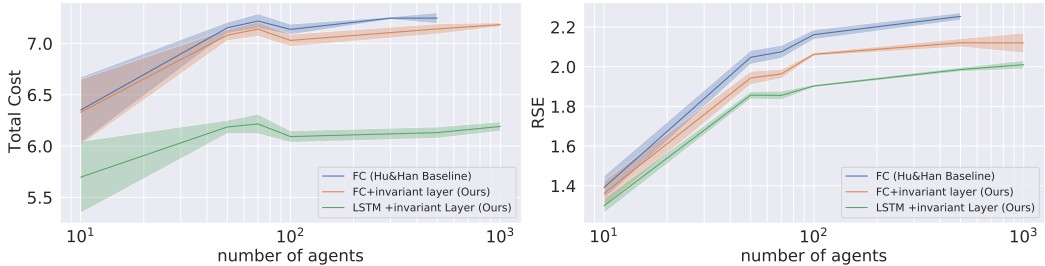

Figure 5: RSE and total loss comparison between our FBSDE framework and that of baseline Han & Hu (2019)

experiments hereon correspond to the symmetric SDFP implementation in Algorithm 1 We also test sample efficiency and generalization capability of the invariant layer on a 50-agent problem trained over 100 stages. The number of initial states is limited during the training and the evaluation criterion is the terminal cost of the test set in which the initial states are different from the initial states during training. Fig. 2 showcases the improvement in sample efficiency and generalization performance of invariant layer. We suspect this is due to the network needing to learn with respect to a specific permutation of the input, whereas permutation invariance is built into the network architecture with invariant layer.

**Importance sampling:** An important distinction of SDFP from the baseline in Han & Hu (2019) is the importance sampling scheme, which helps the LSTM architecture achieve a fast convergence rate during training. However, the baseline, which uses fully connected layer as backbone, is not suitable for importance sampling, as it would lead to an extremely deep network with fully connected layers from gradient topology perspective. Sandler et al. (2018) mentioned that the information loss is inevitable for this kind of fully connected deep network with nonlinear activation. On the other hand, LSTM does not suffer from this problem because of the existence of long and short memory. We illustrate the benefits of importance sampling for LSTM backbone and gradient flow of fully connected layer backbone in Appendix I.

**High dimension experiment:** We also analyze the performance of our framework and that of Han & Hu (2019) both with and without invariant layer on high dimensional problems. We first demonstrate the mitigation of the invariant layer on the curse of many agents. Fig. 5 demonstrates the ablation experiment of the two deep FBSDE frameworks (SDFP and Han & Hu (2019)). In order to illustrate that invariant layer can mitigate the curse of dimensionality, we also integrate invariant layers on Han & Hu (2019) and shows the performance in the same figure 5. In this experiment, the weights of FBSDE frameworks with invariant layer are adjusted in order to dismiss the performance improvement resulting from increased weights from the invariant layers. The total cost and RSE are computed by averaging the corresponding values over the last twenty stages of each run. It can be observed from the plot that without invariant layer, the framework suffers from curse of many agents in the prediction of initial value as the RSE increases with respect to the number of agents. On the other hand, RSE increases at a slower rate with invariant layer. In terms of total cost, which is computed from the cost function defined in eq.2, our framework enjoys the benefits of importance sampling and invariant layer, and achieves better numerical results over the number of agents. We further analyse the influence of invariant layers in the training phase by demonstrating fig.7. Invariant layer helps mintage over-fitting phenomenon in the training and evaluation phase, meanwhile accelerating the training process, even though both of frameworks adopting same feature extracting backbone (LSTM) and importance sampling technique.

We also show that the invariant layer accelerates training empirically on a 500-agent problem. Fig. 6 shows that the FBSDE frameworks converge much faster with invariant layer than without it. We suspect that the acceleration effect results from increased sample efficiency. Note that the comparison is done on a 500-agent problem because the framework does not scale to 1000 agents without the invariant layer. A comparison of the two frameworks with invariant layer only on a 1000-agent problem can be found in Fig 16, which shows similar results to the 500-agent problem.

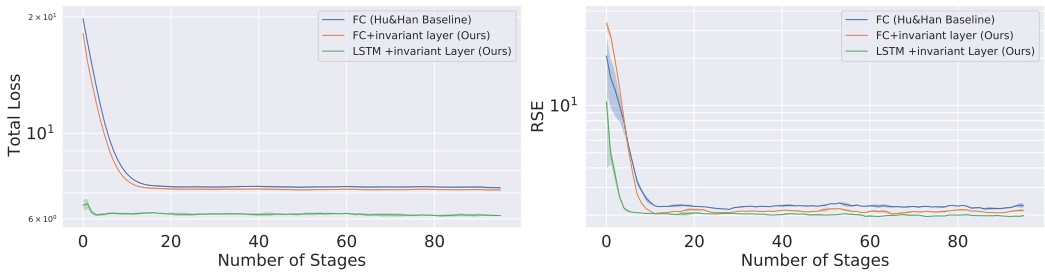

Figure 6: RSE and total loss trajectory comparison between our FBSDE framework and that of Han & Hu (2019) w/ and w/o invariant layer for 500 agents.

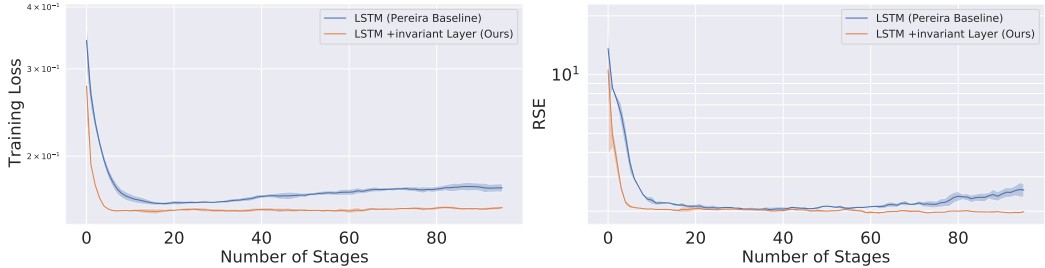

Figure 7: RSE and training loss trajectory comparison between our FBSDE framework and the extension of Pereira et al. (2019) w/ and w/o invariant layer for 500 agents.

**Superlinear Simulation:** We also consider a variant of dynamics in section 4.1,

$$\mathrm{d}X_t^i = \left[a(\bar{X} - X_t^i)^3 + u_t^i\right]\mathrm{d}t + \sigma(\rho\mathrm{d}W_t^0 + \sqrt{1-\rho^2}\mathrm{d}W_t^i), \bar{X}_t = \frac{1}{N}\sum_{i=1}^N X_t^i, i \in \mathbb{I}. \quad (9)$$

Due to the nonlinearity in the drift term, analytic solution or simple numerical representation of the Nash equilibrium does not exist (Han & Hu, 2019). The drift rate $a$ is set to $1.0$ to compensate for the vanishing drift term caused by super-linearity. Heuristically, the the distribution of control and state should be more concentrated than that of the linear dynamics. We compare the state and control of a fixed agent $i$ at terminal time against analytic solution and deep FBSDE solution of the linear dynamics with the same coefficients. Fig. 8 is generated by evaluating the trained deep FBSDE model with a batch size of 50000. It can be observed that the solution from super-linear dynamics is more concentrated as expected. The terminal control distribution plot verifies that the super-linear drift term pushes the state back to the average faster than linear dynamics and thus requires less control effort. Since the numerical solution is not available in the superlinear case, we compare the total loss and training loss between baseline Han & Hu (2019) and our algorithm in the appendix 12.

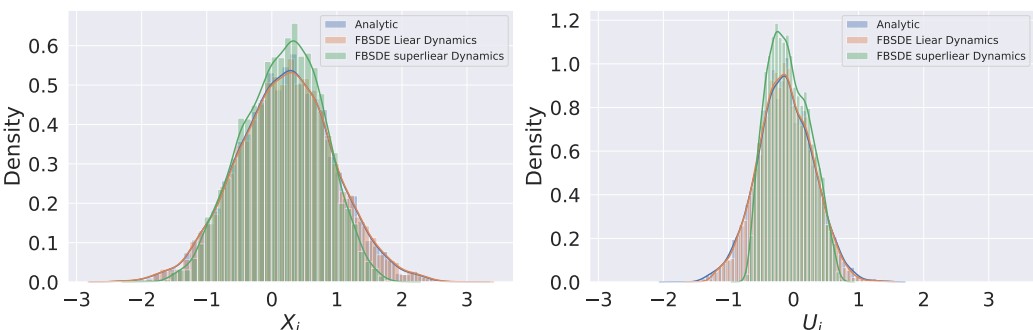

Figure 8: Terminal time step state $\boldsymbol{X}$ and control $\boldsymbol{U}$ distribution of $i$th agent for linear and superlinear dynamics.

### 4.2 BELIEF SPACE AUTONOMOUS RACING

In this section, we demonstrate the general applicability of our framework on an autonomous racing example in belief space. We consider a 2-car autonomous racing competition problem with racecar dynamics

$$\dot{\mathbf{x}} = [v\cos\theta, v\sin\theta, u_{\text{acc}} - c_{\text{drag}}v, u_{\text{steer}}v/L]^{\text{T}} \qquad (10)$$

where $\mathbf{x} = [x, y, v, \theta]^{\text{T}}$ represent the x, y position, forward velocity and heading respectively. Here we assume $x, y, v, u_{\text{acc}} \in \mathbb{R}, u_{\text{steer}} \in [-1, 1]$. The goal of each player is to drive faster than the opponent, stay on the track and avoid collision. An additional competition loss can be added to facilitate competition between players. During the competition, players have access to the global augmented states and opponent's history controller. Additionally, we assume that stochasticity enters the system through the control channels and have a continuous-time noisy observation model with full-state observation. The FBSDE derivation of belief space stochastic dynamics is included in Appendix G.

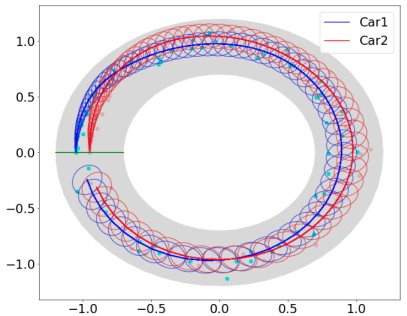

The framework for the racing problem is trained with batch size of 64, and 100 time steps over a time horizon of 10 seconds. Since all the trials will run over 1 lapse of the circle, here we only show the first 8 second result for neatness. Fig. 13 demonstrate the capability of our framework. When there is no competition loss, both of cars can stay in the track. Since there is no competition between two cars, they demonstrate similar behaviors. When we add competition loss on both cars, both of them try to cut the corner in order to occupy the leading position as shown in the second plot in Fig. 13. If competition loss is present in only one of the two cars, then the one with competition loss will dominate the game as shown in the botton subplots of Figure 13. Notably the simulation is running in belief space where all states are estimated with

Figure 9: One belief space racing trajectory. The solid line represents the mean and the circles represent the variance.

observation noise and additive noise in the system. The results emphasizes the generalization ability of our framework on more complex systems with higher state and control dimensions. Fig. 9 shows a single trajectory of each car's posterior distribution.

## 5 CONCLUSION

In this paper, we propose a scalable deep learning framework for solving multi-agent stochastic differential game using fictitious play. The framework relies on the FBSDE formulation with importance sampling for sufficient exploration. In the symmetric game setup, an invariant layer is incorprated to render the framework agnostic to permutatoon of agents and further reduce the memory complexity. The scalability of this algorithm, along with a detailed sensitivity analysis, is demonstrated in an inter-bank borrowing/lending example. The framework achieves lower loss and scales to much higher dimensions than the state of the art. The general applicability of the framework is showcased on a belief space autonomous racing problem in simulation.

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

APPENDIX

## A MULTI-AGENT HJB DERIVATION

Applying Bellman's principle to the value function equation 3 as following

$$
\begin{aligned}
V^i(t, \boldsymbol{X}(t)) &= \inf_{\boldsymbol{u}_i \in \mathbb{U}_i} \mathbb{E}\left[V^i(t + \mathrm{d}t, \boldsymbol{X}(t+dt)) + \int_t^{t+dt} C^i \mathrm{d}\tau\right] \\
&= \inf_{\boldsymbol{u}_i \in \mathbb{U}_i} \mathbb{E}\Big[C^i \mathrm{d}t + V^i(t, \boldsymbol{X}(t)) + V_t^i(t, \boldsymbol{X}(t))\mathrm{d}t \\
&\quad + V_x^{i\mathrm{T}}(t, \boldsymbol{X}(t))\mathrm{d}\boldsymbol{X} + \frac{1}{2}\mathrm{tr}(V_{xx}(t, \boldsymbol{X}(t)\Sigma\Sigma^{\mathrm{T}})\mathrm{d}t\Big] \\
&= \inf_{\boldsymbol{u}_i \in \mathbb{U}_i} \mathbb{E}\Big[C^i \mathrm{d}t + V^i(t, \boldsymbol{X}(t)) + V_t^i(t, \boldsymbol{X}(t))\mathrm{d}t \\
&\quad + V_x^{i\mathrm{T}}(t, \boldsymbol{X}(t))((f + G\boldsymbol{U})\mathrm{d}t + \Sigma\mathrm{d}\boldsymbol{W}) + \frac{1}{2}\mathrm{tr}(V_{xx}^i(t, \boldsymbol{X}(t))\Sigma\Sigma^{\mathrm{T}})\mathrm{d}t\Big] \\
&= \inf_{\boldsymbol{u}_i \in \mathbb{U}_i} \Big[C^i \mathrm{d}t + V^i(t, \boldsymbol{X}(t)) + V_t^i(t, \boldsymbol{X}(t))\mathrm{d}t \\
&\quad + V_x^{i\mathrm{T}}(t, \boldsymbol{X}(t))((f + G\boldsymbol{U})\mathrm{d}t) + \frac{1}{2}\mathrm{tr}(V_{xx}^i(t, \boldsymbol{X}(t))\Sigma\Sigma^{\mathrm{T}})\mathrm{d}t\Big] \\
\Rightarrow 0 &= V_t^i(t, \boldsymbol{X}(t)) + \inf_{\boldsymbol{u}_i \in \mathbb{U}_i}\left[C^i + V_x^{i\mathrm{T}}(t, \boldsymbol{X}(t))(f + G\boldsymbol{U})\right] + \frac{1}{2}\mathrm{tr}(V_{xx}^i(t, \boldsymbol{X}(t))\Sigma\Sigma^{\mathrm{T}})
\end{aligned}
\tag{11}
$$

Given the cost function assumption, the infimum can be obtained explicitly using optimal control $\boldsymbol{u}_{i,m}^* = -R^{-1}(G_i^{\mathrm{T}}V_x^i + Q_i^{\mathrm{T}}x)$. With that we can obtain the final form of the HJB PDE as

$$
V_t^i + h + V_x^{i\mathrm{T}}(f + G\boldsymbol{U}_{0,-i}) + \frac{1}{2}\mathrm{tr}(V_{xx}^i\Sigma\Sigma^{\mathrm{T}}) = 0, \quad V^i(T, \boldsymbol{X}) = g(\boldsymbol{X}(T)). \tag{12}
$$

## B FBSDE DERIVATION

Given the HJB PDE in equation 4, one can apply the nonlinear Feynman-Kac lemma Han & Hu (2019) to obtain a set of FBSDE as

$$
\begin{aligned}
\mathrm{d}\boldsymbol{X}(t) &= (f + G\boldsymbol{U}_{0,-i})\mathrm{d}t + \Sigma\mathrm{d}\boldsymbol{W}, \quad \boldsymbol{X}(0) = \boldsymbol{x}_0 \\
\mathrm{d}V^i &= -h\mathrm{d}t + V_x^{i\mathrm{T}}\Sigma\mathrm{d}W, \quad V(\boldsymbol{X}(T)) = g(\boldsymbol{X}(T)).
\end{aligned}
\tag{13}
$$

Note that the forward process $\boldsymbol{X}$ is driven by the control of all agents other than $i$. This means that agent $i$ searches the state space with Brownian motion only to respond to other agents' strategies. To increase the efficiency of the search, one can add any control from agent $i$ to guide its exploration, as long as the backward process is compensated for accordingly. In this work, since we consider problems with a closed form solution of the optimal control $\boldsymbol{u}_{i,m}$, we add it to the forward process for importance sampling from a new set of FBSDEs.

$$
\begin{aligned}
\mathrm{d}\boldsymbol{X} &= (f + G\boldsymbol{U}_{*,-i})\mathrm{d}t + \Sigma\mathrm{d}\boldsymbol{W}, \quad \boldsymbol{X}(0) = \boldsymbol{x}_0 \\
\mathrm{d}V^i &= -(h + V_x^{i\mathrm{T}}G\boldsymbol{U}_{*,0})\mathrm{d}t + V_x^{\mathrm{T}}\Sigma\mathrm{d}W, \quad V(T) = g(\boldsymbol{X}(T)).
\end{aligned}
\tag{14}
$$

## C CONTINUOUS TIME EXTENDED KALMAN FILTER

The Partial Observable Markov Decision Process is generally difficult to solve within infinite dimensional space belief. Commonly, the Value function does not have explicit parameterized form. Kalman filter overcome this challenge by presuming the noise distribution is Gaussian distribution. In order to deploy proposed Forward Backward Stochastic Differential Equation (FBSDE) model in the Belief space, we need to utilize extended Kalman filter in continuous time Jazwinski (1970) correspondingly. Given the partial observable stochastic system:

$$
\frac{\mathrm{d}x}{\mathrm{d}t} = f(x, u, w, t), \quad \text{and} \quad z = h(x, v, t) \tag{15}
$$

Where $f$ is the stochastic state process featured by a Gaussian noise $w \sim \mathcal{N}(0, Q)$, $h$ is the observation function while $v \sim \mathcal{N}(0, R)$ is the observation noise. Next, we consider the linearization of the stochastic dynamics in equation 20 represented as follows:

$$A = \frac{\partial f}{\partial x}\Big|_{\hat{x}}, L = \frac{\partial f}{\partial w}\Big|_{\hat{x}}, C = \frac{\partial h}{\partial x}\Big|_{\hat{x}}, M = \frac{\partial h}{\partial v}\Big|_{\hat{x}}, \tilde{Q} = LQL^{\mathrm{T}}, \tilde{R} = MRM^{\mathrm{T}} \qquad (16)$$

one can write the posterior mean state $\hat{x}$ and prior covariance matrix $P^-$ estimation update rule by Simon (2006):

$$\begin{aligned}
\hat{x}(0) &= \mathbb{E}[x(0)], \quad P^-(0) = \mathbb{E}[(x(0) - \hat{x})(x(0) - \hat{x})^{\mathrm{T}}] \\
K &= PC^{\mathrm{T}}\tilde{R}^{-1} \\
\dot{\hat{x}} &= f(\hat{x}, u, w_0, t) + K[z - h(\hat{x}, v_0, t)] \\
\dot{P}^- &= AP^- + P^- A^{\mathrm{T}} + \tilde{Q} - P^- C^{\mathrm{T}}\tilde{R}^{-1}CP^-
\end{aligned} \qquad (17)$$

We follow the notation in (Simon, 2006), where $x$ is the real state, $\hat{x}$ is the mean of state estimated by Kalman filter based on the noisy sensor observation, $P^-$ represents for the covariance matrix of the estimated state, nominal noise values are given as $w_0 = 0$ and $v_0 = 0$, where superscript + is the posterior estimation and − is the prior estimation. Then we can define a Gaussian belief dynamics as $\boldsymbol{b}(\hat{x}_k, P_k^-)$ by the mean state $\hat{x}$ and variance $P^-$ of normal distribution $\mathcal{N}(\hat{x}_k, P_k^-)$

The belief dynamics results in a decoupled FBSDE system as follows:

$$\begin{aligned}
\mathrm{d}\boldsymbol{b}_k &= g(\boldsymbol{b}_k, \boldsymbol{u}_k, 0)\mathrm{d}t + \Sigma(\boldsymbol{b}_k, \boldsymbol{u}_k, 0)\mathrm{d}W, \mathrm{d}W \sim \mathcal{N}(0, I) \\
\mathrm{d}V &= -C^{i^\star}\mathrm{d}t + V_x^{i\mathrm{T}}\Sigma\mathrm{d}W
\end{aligned} \qquad (18)$$

where:

$$\begin{aligned}
g(\mathbf{b}_k, \mathbf{u}_k) &= \begin{bmatrix} b(t, \boldsymbol{X}(t), \boldsymbol{u}_{i,m}(t); \boldsymbol{u}_{-i,m}) \\ vec(A_k P_k^- + P_k^- A_k^{\mathrm{T}} + \tilde{Q}_k - P_k^- C_k^{\mathrm{T}}\tilde{R}_k^{-1}C_k P_k^-) \end{bmatrix} \\
\Sigma(\mathbf{b}_k, \mathbf{u}_k) &= \begin{bmatrix} \sqrt{K_k C_k P_k^-}\,\mathrm{d}t \\ \mathbf{0} \end{bmatrix} \\
V(T) &= g(\boldsymbol{X}(T)) \\
\hat{\boldsymbol{X}}(0) &= \mathbb{E}[\boldsymbol{X}(0)] \\
P^-(0) &= \mathbb{E}[(\boldsymbol{X}(0) - \hat{\boldsymbol{X}})(\boldsymbol{X}(0) - \hat{\boldsymbol{X}})^{\mathrm{T}}]
\end{aligned} \qquad (19)$$

## D   DEEP SETS

A function $f$ maps its domain from $\mathcal{X}$ to $\mathcal{Y}$. Domain $\mathcal{X}$ is a vector space $\mathbb{R}^d$ and $\mathcal{Y}$ is a continuous space $\mathbb{R}$. Assume the function take a set as input: $\mathbb{X} = \{x_1...x_N\}$, then the function $f$ is indifferent if it satisfies property (Zaheer et al., 2017).

**Property 1.** A function $f : \mathcal{X} \to \mathcal{Y}$ defined on sets is permutation invariant to the order of objects in the set. i.e. For any permutation function $\pi$: $f(\{x_1...x_N\}) = f(\{x_{\pi(1)}...x_{\pi(N)}\})$

In this paper, we discuss when $f$ is a nerual network strictly.

**Theorem 1** $\boldsymbol{X}$ *has elements from countable universe. A function* $f(\boldsymbol{X})$ *is a valid permutation invariant function, i.e invariant to the permutation of* $\boldsymbol{X}$*, iff it can be decomposed in the from* $\rho(\sum_{x \in \boldsymbol{X}} \phi(x))$*, for appropriate function* $\rho$ *and* $\phi$*.*

In the symmetric multi-agent system, each agents is not distinguishable. This property gives some hints about how to extract the features of $-i$th agents by using neural network. The states of $-i$th agents can be represented as a set: $\boldsymbol{X} = \{X_1, X_2, ..., X_{i-1}, X_{i+1}, ..., X_N\}$. We want to design a neural network $f$ which has the property of permutation invariant. Specifically, $\phi$ is represented as a one layer neural network and $\rho$ is a common nonlinear activation function.

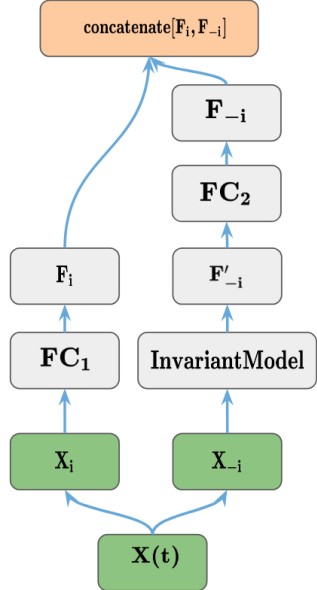

Figure 10: Invariant layer architecture.

# E  INVARIANT LAYER ARCHITECTURE

The architecture of invariant layer is described in Fig. 10. The input of the layer is the states at time step $t$. The Invariant Model module in Fig. 10 is described in Appendix D, where $\phi$ is a neural network and $\rho$ is nonlinear activation function. The specific configuration of neural network in Invariant model can be found in J.

Noticing that all the agents has the access to the global states, we define the state input features of neural network for $i$th agent as:

$$\boldsymbol{X}_{t,i} = \{x_i, x_1, x_2..., x_{i-1}, x_{i+1}, ...x_N\}, \tag{20}$$

with shape of $[BS, N]$. In the other word, we always put own feature at first place. For each agent $i$, there exists such feature tensor, then the shape of input tensor will become $[BS, N, N]$ for invariant layer. In invariant layer, we first separate the input feature $\boldsymbol{X}_t$ into two parts: $\boldsymbol{X}_{t,i}$ and $\boldsymbol{X}_{t,-i}$. Then the features of $-i$th agents $\boldsymbol{X}_{t,-i}$ will be sent to the invariant model. The shape of $\boldsymbol{X}_{t,-i}$ will be $[B, N, N-1]$ where $N$ is the number of agents. First,we could use neural network to map the feature into $N_f$ dimension space, where $N_f$ is the feature dimensions. Then the shape of the tensor will become $[BS, N, N-1, N_f]$, After summing up the features of all the element in the set, the dimension of the tensor would reduce to $[BS, N, 1, N_f]$, and we denote this feature tensor as $F_1$. However, the memory complexity is $\mathcal{O}(N^2 \times N_f)$ which is not tolerable when the number of agent $N$ increases. Alternatively, we can simply mapping the feature tensor $[BS, N]$ into desired feature dimension $N_f$, then the tensor would become $[BS, N, N_f]$, and we denote it to be $F_2$. Now we create another tensor which is the average of features of element in set with size $[BS, 1, N_f]$ and we denote it to be $\bar{F}_2$. Then we denote $F_2' = (\bar{F}_2 \times N - F_2)/(N-1)$ which has size of $[BS, N, N_f]$. We can find that $F_2' = F_1$, and the memory complexity of computing $F_2'$ is just $\mathcal{O}(N)$. The derivation is true if the system is symmetric and the agents are not distinguishable. The trick can be extended to high state dimension for individual agent.

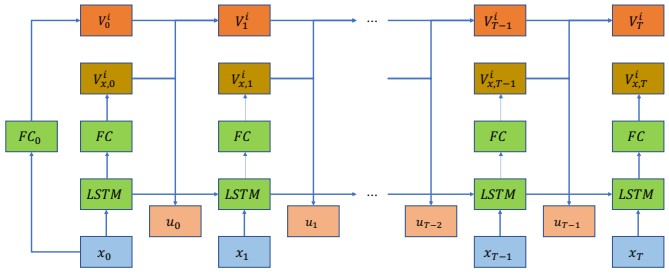

Figure 11: FBSDE Network for a Single Agent. Note that the same FC is shared across all timesteps.

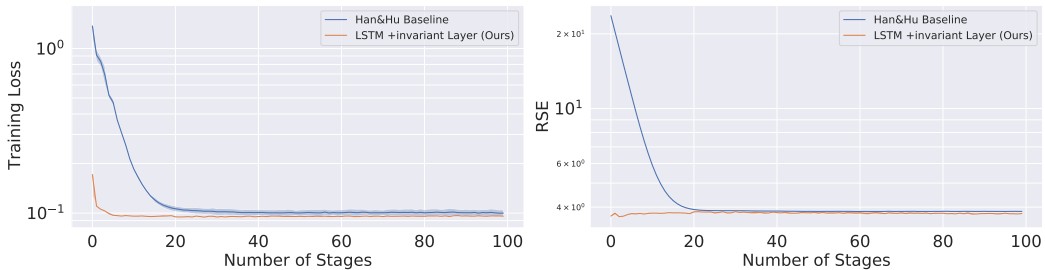

Figure 12: Superlinear dynamics comparison between baseline and our algorithm

## F INTERBANK

By pluging the running cost to the HJB function, one can have,

$$
V_{i,t} + \inf_{\boldsymbol{u}_i \in \mathbb{U}_i} \left[ \sum_{j=1}^{N} [a(\bar{\boldsymbol{X}} - \boldsymbol{X}_j) + u_j^2] V_{x_j} + \frac{1}{2}\boldsymbol{u}_i^2 - q\boldsymbol{u}_i(\bar{\boldsymbol{X}} - \boldsymbol{X}_i) + \frac{\epsilon}{2}(\bar{\boldsymbol{X}} - \boldsymbol{X}_i)^2 \right] +
$$
$$
\frac{1}{2}\mathrm{tr}(V_{xx,i}\Sigma\Sigma^{\mathrm{T}}) = 0. \tag{21}
$$

By computing the infimum explicitly, the optimal control of player $i$ is: $\boldsymbol{u}_i(\boldsymbol{X}, t) = q(\bar{\boldsymbol{X}} - \boldsymbol{X}_i) - V_{x,i}(\boldsymbol{X}, t)$. The final form of HJB can be obtained as

$$
V_{i,t} + \frac{1}{2}\mathrm{tr}(V_{xx,i}\Sigma\Sigma^{\mathrm{T}}) + a(\bar{\boldsymbol{X}} - \boldsymbol{X}_i)V_{x,i} + \sum_{j \neq i}[a(\bar{\boldsymbol{X}} - \boldsymbol{X}_j) + \boldsymbol{u}_j]V_{x,j}
$$
$$
+ \frac{\epsilon}{2}(\bar{\boldsymbol{X}} - \boldsymbol{X}_i)^2 - \frac{1}{2}(q(\bar{\boldsymbol{X}} - \boldsymbol{X}_i) - V_{x,i})^2 = 0 \tag{22}
$$

Applying Feynman-Kac lemma to equation 22, the corresponding FBSDE system is

$$
\mathrm{d}\boldsymbol{X}(t) = (f(\boldsymbol{X}(t), t) + G(\boldsymbol{X}(t), t)\boldsymbol{u}(t))\mathrm{d}t + \Sigma(t, \boldsymbol{X}(t))\mathrm{d}\boldsymbol{W_t}, \quad \boldsymbol{X}(0) = \boldsymbol{x}_0
$$
$$
\mathrm{d}V_i = -[\frac{\epsilon}{2}(\bar{\boldsymbol{X}} - \boldsymbol{X}_i)^2 - \frac{1}{2}(q(\bar{\boldsymbol{X}} - \boldsymbol{X}_i) - V_{x,i})^2 + \boldsymbol{u}_i]\mathrm{d}t + V_{x_i}^{\mathrm{T}}\Sigma dW, \quad V(T) = g(\boldsymbol{X}(T)). \tag{23}
$$

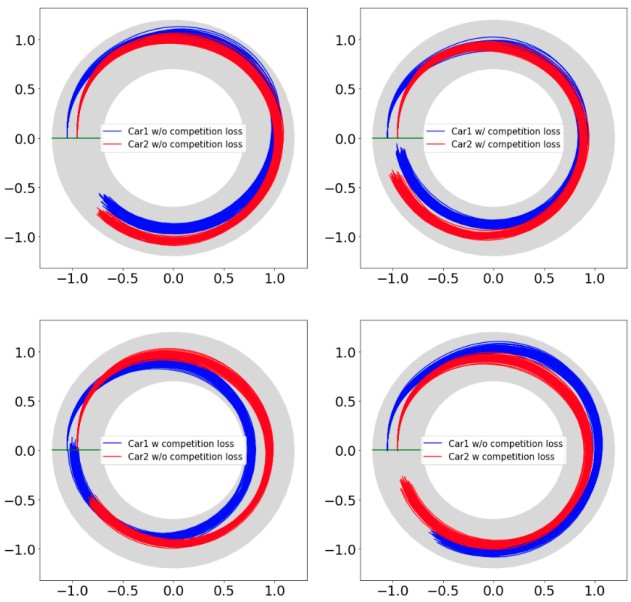

Figure 13: 2 car racing problem with 8 second time horizon.

## G  BELIEF CAR RACING

The full stochastic model can be written as

$$d\mathbf{x} = (f(\mathbf{x}) + G(\mathbf{x})\mathbf{u})dt + \Sigma(\mathbf{x})dw, \quad \mathbf{z} = h(\mathbf{x}) + m$$

$$f(\mathbf{x}) = \begin{bmatrix} v\cos\theta \\ v\sin\theta \\ -c_{\text{drag}}v \\ 0 \end{bmatrix}, \quad G(\mathbf{x}) = \Sigma(\mathbf{x}) = \begin{bmatrix} 0 & 0 \\ 0 & 0 \\ 1 & 0 \\ 0 & v/L \end{bmatrix}, \quad h(\boldsymbol{x}) = \boldsymbol{x} \tag{24}$$

Where $dw$ is standard brownian motion. We consider the problem of two cars racing on circle track. The cost function of each car is designed as

$$J_t = \underbrace{\exp\left(\left|\frac{x^2}{a^2} + \frac{y^2}{b^2} - 1\right|\right)}_{\text{track cost}} + \underbrace{\text{ReLU}\left(-v\right)}_{\text{velocity cost}} + \underbrace{\exp\left(-d\right)}_{\text{collision cost}}$$

Where $d$ is Euclidean distance between two cars. In this showcase, we use continuous time extended Kalman Filter to propagate belief space dynamics described in equation 19. The detailed algorithm for Belief space deep fictitious play FBSDE can be found in Appendix.

We introduce the concept of game by using an additional competition cost:

$$J_{competition} = \exp(-\begin{bmatrix} cos(\theta) \\ sin(\theta) \end{bmatrix}^{\mathrm{T}} \begin{bmatrix} x_1 - x_2 \\ y_1 - y_2 \end{bmatrix})$$

Where $x_i, y_i$ is the $x, y$ position of $i$th car. When $i$th car is leading, the competition loss will be minor, and it will increase exponentially when the car is trailing.

Thanks to decoupled BSDE structure, each car can measure this competition loss separately and optimize the value function individually.

## H  ANALYTIC SOLUTION FOR INTER-BANK BORROWING/LENDING PROBLEM

The analytic solution for linear inter-bank problem was derived in Carmona et al. (2013). We provide them here for completeness. Assume the ansatz for HJB function is described as:

$$V_i(t, \boldsymbol{X}) = \frac{\eta(t)}{2}(\bar{\boldsymbol{X}} - \boldsymbol{X}_i)^2 = \mu(t) i \in \mathbb{I} \tag{25}$$

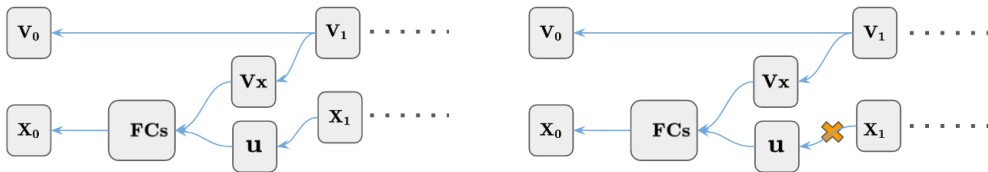

Figure 14: The gradient path of FBSDE model w/ and w/o importance sampling. The figure on the left is FBSDE with importance sampling and the figure on the right is FBSDE without importance sampling. One can identify that the framework with importance sampling would lead to long chain of gradient.

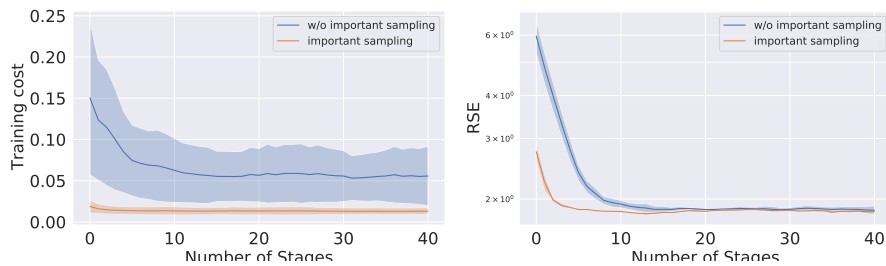

Figure 15: The training loss and RSE of LSTM backbone FBSDE architecture w/ and w/o importance sampling

Where $\eta(t), \mu(t)$ are two scalar functions. The optimal control under this ansatz is:

$$\alpha_i^\star(t, \boldsymbol{X}) = \left[q + \eta(t)(1 - \frac{1}{N})\right](\bar{\boldsymbol{X}} - \boldsymbol{X}_i) \tag{26}$$

By pluginging the ansatz into HJB function derived in 22, one can have,

$$\dot{\eta}(t) = 2(a + q)\eta(t) + (1 - \frac{1}{N^2})\eta^2(t) - (\epsilon - q^2), \eta(T) = c,$$
$$\dot{\mu}(t) = -\frac{1}{2}\sigma^2(1 - \rho^2)(1 - \frac{1}{N})\eta(t), \mu(T) = 0. \tag{27}$$

There exists the analytic solution for the Riccati equation described above as,

$$\eta(t) = \frac{-(\epsilon - q^2)(e^{(\delta^+ - \delta^-)(T-t)} - 1) - c(\delta^+ e^{(\delta^+ - \delta^-)(T-t)} - \delta^-)}{(\delta^- e^{(\delta^+ - \delta^-)(T-t)} - \delta^+) - c(1 - 1/N^2)(e^{(\delta^+ - \delta^-)(T-t)}) - 1}. \tag{28}$$

Where $\delta^\pm = -(a + q) \pm \sqrt{R}$ and $R = (a + q)^2 + (1 - 1/N^2)(\epsilon - q^2)$

## I    IMPORTANCE SAMPLING

Fig. 14 demonstrates how fully connected layers with importance sampling would lead to a extreme deep fully connected neural network. Fig. 15 demonstrates how importance sampling helps increase convergence rate in FBSDE with LSTM backbone. The experiment is conducted with 50 agents and 50 stages. All the configuration is identical except the existence of importance sampling.

## J    EXPERIMENT CONFIGURATIONS

This Appendix elaborates the experiment configurations for section 4. For all the simulation in section 4, the number of SGD iteration is fixed as $N_{SGD} = 100$. We are using Adam as optimizer

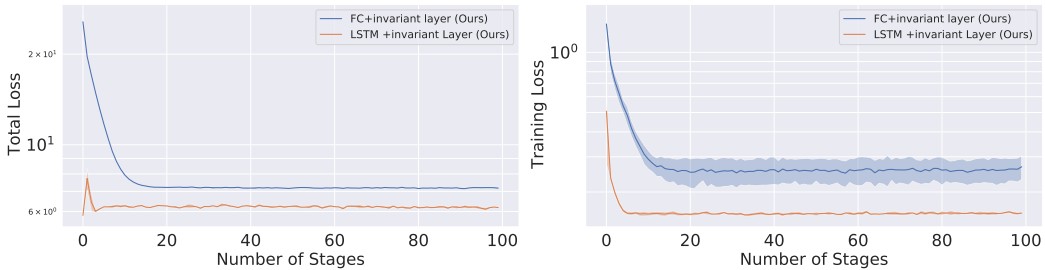

Figure 16: The Total loss and RSE of 1000 agents simulation

with 1E-3 learning rate for all simulations.

In section 4.1, For the prediction of initial value function, all of frameworks are using 2 layers feed forward network with 128 hidden dimension. For the baseline framework, we followed the suggested configuration motioned in Han et al. (2018). At each time steps, $V_{x,i}$ is approximated by three layers of feed forward network with 64 hidden dimensions. We add batch norm Ioffe & Szegedy (2015) after each affine transformation and before each nonlinear activation function. For Deep FBSDE with LSTM backbone, we are using two layer LSTM parametrized by 128 hidden state. If the framework includes the invariant layer, the number of mapping features is chosen to be 256. The hyperparameters of the dynamics is listed as following:

$$a = 0.1, \ q = 0.1, \ c = 0.5, \ \epsilon = 0.5, \ \rho = 0.2, \ \sigma = 1, T = 1. \tag{29}$$

In the simulation, the time horizon is separated into 40 time-steps by Euler method. Learning rate is chosen to be 1E-3 which is the default learning rate for Adam optimizer. The initial state for each agents are sampled from the uniform distribution $[\delta_0, \delta_1]$. Where $\delta_0$ is the constant standard deviation of state $\boldsymbol{X}(t)$ during the process. In the evaluation, we are using 256 new sampled trajectory which are different from training trajectory to evaluate the performance in RSE error and total cost error. The number of stage is set to be 100 which is enough for all framework to converge.

In section 4.2, the hyperparameter is listed as following:

$$c_{drag} = 0.01, \ L = 0.1, \ c = 0.5, \ T = 10.0 \tag{30}$$

The observation noise is sampled from Gaussian noise $m \sim \mathcal{N}(0, 0.01\boldsymbol{I})$. The time horizon is enrolled into 100 time-steps by Euler method. In this experiments, the initial value $V_i$ is approximated a single trainable scale and $V_{x,i}(t)$ is approximated by two layers of LSTM parametrized with 32 hidden dimensions. The number of stage is set to be 10.

---

**Algorithm 2** Scalable Deep Fictitious Play FBSDE

---

1: **Hyper-parameters**:$N$: Number of players, $T$: Number of timesteps, $M$: Number of stages in fictitious play, $N_{gd}$: Number of gradient descent steps per stage, $\boldsymbol{U}_0$: the initial strategies for players in set $\mathbb{I}$, $B$: Batch size, $\epsilon$: training threshold, $\Delta t$: time discretization
2: **Parameters**:$V(\boldsymbol{x}_0; \phi)$: Network weights for initial value prediction, $\theta$: Weights and bias of fully connected layers and LSTM layers.
3: Initialize trainable papermeters:$\theta^0$, $\phi^0$
4: **while** LOSS is above certain threshold $\epsilon$ **do**
5:    **for** $m \leftarrow 1$ to $M$ **do**
6:       **for** all $i \in \mathbb{I}$ *in parallel* **do**
7:          Collect opponent agent's policy $f^{m-1}_{LSTM_{-i}}(\cdot), f^{m-1}_{FC_{-i}}(\cdot)$
8:          **for** $l \leftarrow 1$ to $N_{gd}$ **do**
9:             **for** $t \leftarrow 1$ to $T-1$ **do**
10:                **for** $j \leftarrow 1$ to $B$ *in parallel* **do**
11:                   Compute network prediction for $i$th player: $V^i_{x_i,j,t} = f^m_{FC_i}(f^m_{LSTM_i}(\boldsymbol{X}^j_t; \theta^{l-1}_i))$
12:                   Compute $i$th optimal Control:$\boldsymbol{u}^{j,\star}_{i,t} = -R^{-1}_i(G^{\mathrm{T}}_i V^i_{x_i,j,t} + Q^{\mathrm{T}}_i \boldsymbol{x}^j_i)$
13:                   Infer $-i$th players' network prediction: $V^i_{x_{-i},j,t} = f^{m-1}_{FC_{-i}}(f^{m-1}_{LSTM_{-i}}(\boldsymbol{X}_t; \theta_{-i}))$
14:                   Compute $-i$th optimal Control:$\boldsymbol{u}^{j,\star}_{-i,t} = -R^{-1}_{-i}(G^{\mathrm{T}}_{-i} V^j_{x_{-i},t} + Q^{\mathrm{T}}_{-i} \boldsymbol{x}^j_{-i})$
15:                   Sample noise $\Delta W^j \sim \mathcal{N}(0, \Delta t)$
16:                   Propagate FSDE: $\boldsymbol{X}^j_{t+1} = f_{FSDE}(\boldsymbol{X}^j_t, \boldsymbol{u}^{j,\star}_{i,t}, \boldsymbol{u}^{j,*}_{-i,t}, \Delta W^j, t)$
17:                   Propagate BSDE: $V^i_{j,t+1} = f_{BSDE}(V^i_{j,t}, \boldsymbol{X}^j_t, \boldsymbol{u}^{j,\star}_{i,t}, \Delta W^j, t)$
18:                **end for**
19:             **end for**
20:             Compute loss: $\mathcal{L} = \frac{1}{B} \sum_{j=1}^{B} (V^{i,\star}_{j,T} - V^i_{j,T})^2$
21:             Gradient Update: $\theta^l, \phi^l$
22:          **end for**
23:       **end for**
24:    **end for**
25: **end while**

---

