# OpenReview forum: "Multi-agent Deep FBSDE Representation For Large Scale  Stochastic Differential Games"
_ICLR.cc/2021/Conference — Reject_

### Official Review · AnonReviewer1 · 2020-10-27
**Puts together prior works and exploits problem structure to achieve scaling. Yet, critical details related to scaling are missing.**

**Rating:** 5
**Confidence:** 4

**Review:**

Clarity:
Till page 3 the paper was easy to follow, i.e., the analytical expressions in eq(5), and the basic idea of Algorithm 1 (which is same as prior works by Han et al. , Wang et al., Periera et al.) are clear.  However, after page 3 the paper is hard to follow. The specific points are as follows:
1) It looks like both LSTM and FC have been tried in the past.  So what is the novelty here? Is it making the deep FBSDE framework scalable?  and if so how is this achieved? by introducing the invariant layer? or just using the symmetry property to share the networks?
2) Comparisons with Han& Hu (2019) is presented only in Table 2. Figure 5 says comparison with Han & Hu (2019), but only FC w/wo invariant layer and LSTM w/wo invariant layer have been presented. Is it true that Han & Hu (2019) is FC and LSTM without the invariant layer and the current approach in the paper is FC and LSTM with the invariant layer?
3) Figure 2, says comparison between analytical and deep FBDSDE solutions. Yet, 10 trajectories are displayed. Do they correspond to 10 different agents? If  so, what is the difference between the 10 trajectories and the deep FBSDE solution?
4) In the autonomous racing section,  the authors  mention "The results emphasizes the generalization ability of our framework" what is generalization here? Is it that the method also works on a different domain?

Quality:
1) The main contribution seems to be scalability. It will be a lot helpful if Section 3.2 is elaborated a bit more, where the `scalability' aspects of the current work is discussed in detail. As of now, Section 3.2 seems `rushed'.
2) Table 2 compares the proposed method with Han&Hu (2019). This is done for 10 agents. Yet, we do not know what happens as we increase the number of agents.
3) Figure 4 has number of agents in x-axis, yet it compares only the various implementations of the current work and prior work is not included.


Novelty:
It appears that the paper puts together blocks/algorithms in prior works (say invariant layer from Zaheer et al. 2017, deep FBSDE from Wang et al., Periera et al., Han & Hu). This way novelty seems  to be limited.

Significance:
Scalability is the main take away of this paper. Figure  5 and Figure 4 show that the proposed method is better than prior methods, yet the performance improvement is not significant. However, the savings could be in the time/space complexity, which the paper does not elaborate. In short, due to the fact that certain critical parts have not been explained it is hard to judge the significance.


Overall Feedback:  The critical information regarding the invariant layers, scaling, savings in the scaling is missing. Also, adding this information may not take significantly extra space. So, it will be great if the authors can provide this information.

---

> ### Author Response · Authors · 2020-11-18
> **Author Response to Reviewer 1**
>
> **Responses for Reviewer 1**:
>
> **1. Novelty and contribution of our work**
>
> The main advantage of the framework introduced is its scalability wrt. time and memory. The use of importance sampling and LSTM architecture accelerate training by improving the sample efficiency and introducing time correlation in the network architecture. The addition of an invariant layer renders our framework invariant to permutations of agents. This also greatly reduces the memory complexity of our framework and improves the performance when the number of agents is large. Our framework scales to 20x the problem in Han &  Hu and achieves better performance on a similar scale.
> Section 3.2 has been updated in the revised paper to provide the details for the scalability aspect.
>
> **2. Updated Figure 5**
>
> We apologize for the confusing legend in the original figure 5. The paper has been updated with a more proper and detailed legend. In short, Han & Hu's work is FC w/o importance sampling, w/o invariant layers. Our approach is LSTM w/ importance sampling and w/ invariant layers. In the experiments, we also artificially integrate the invariant layer with Han&Hu's work to evaluate the generalization and efficiency of the invariant layer. The comparison is also demonstrated in figure 5.
>
> **3. Updated Figure 2**
>
> We apologize for the unclear legend in the original figure 2. The paper has been updated and now the comparison between analytic solution and numerical solution by our FBSDE solver can be found in figure 3. Figure 3’s legend has also been updated to provide a clear distinction between the analytic solution and the proposed framework’s solution. The dot is the analytic solution and the solid line is the numerical solution by our proposed FBSDE solver.
>
> **4. Autonomous racing generalization**
>
> As the reviewer correctly pointed out, the autonomous racing example aims to showcase the general applicability of the framework. Furthermore, in this example, the individual agent’s states and controls are higher dimensional, and the dynamics are more complex than that of the interbank example. (For decoupled BSDE it provides the possibility of application to the competitive scenario. Meanwhile, we also demonstrate that our framework can be applied to partially observed scenarios as long as the forward dynamics can be written into an FSDE form.)

---

> > ### Comment · AnonReviewer1 · 2020-11-24
> > **Thanks for the response.**
> >
> > I hereby acknowledge that I have read the author response.

---

### Official Review · AnonReviewer2 · 2020-10-27
**Some questions about clarity of presentation and significance of the approach**

**Rating:** 4
**Confidence:** 3

**Review:**

Summary

The paper introduces improved deep learning architecture for solving stochastic differential games by fictitious play. Compared to previous best model it uses LSTM instead of MLP to capture forward dynamics of the system, importance sampling and order invariant encoding of other agents to improve sample efficiency of learning.


Strong points

* Order invariant encoding uses inherent structure of the problem that improves sample efficiency compared to previous approaches.
* Batching of some operations speeds up the algorithm.
* Two different domains are used to evaluate the model.


Weak points

* Time complexity claims are problematic.
* The paper is sometimes difficult to follow. It seems like 8 pages aren't enough to describe the whole approach in detail. Many things in appendix seem central for the presented system.
* Importance sampling that is claimed to be important ingredient isn't described well enough neither in the main text nor in the appendix.
* Quantitative evaluation is presented only for domains that can be already solved analytically. In other domains only qualitative results are presented.


Recommendation

I recommend rejecting the paper due to lack of clarity and lack of evaluation against other methods suitable for solving problems that can't be solved analytically.


Questions

* It seems to me that "propagate by batch" time complexities in Table 1 assume constant time computation of batches. Is that correct? That would however be a simplification, while batched computation is much faster on GPUs it isn't in general constant time. This view is further supported by Fig 4 where empirical time complexity also isn't constant as is suggested by Table 1.

* My understanding of empirical results is that the model is run in domains with analytical solution only as sanity check. The true value is in domains without analytical solution (e.g. Superlinear Simulation section). Current evaluation for superlinear setup and autonomous racing is only qualitative. I would like to see comparison to previous methods applicable to these domains if there are any (numerical solvers?). If there aren't any the paper should explain why this is the case. I understand that the model is an improvement over Han & Hu however does it allow us to do something we couldn't do before when we consider even non learning methods? Please reflect on this point to help me better understand main contribution of the paper. This broader context would help readers (like me) that aren't experts in stochastic differential games.


Possible improvements

* Since most of the results compare presented model against Han & Hu 2019 it would help to add a paragraph that briefly summarizes algorithmic/model differences. Is it true that current system is Han & Hu with LSTM instead of MLP + importance sampling + invariant layer?

* In table 2 unit of "Total time" isn't specified.

* "memory complexity of LSTM with respect to time is O(1)" --- this is true for inference time, at train time O(T) activations still have to be stored. (If they aren't to be recomputed again for each t < T). The text can be more clear on that.

* In Figs 5 and 6 captions say that it compares current system against Han & Hu however the legend shows FC and LSTM with and without invariant layers. I assume that Han & Hu = FC and current system is LSTM + invariant layer. However being more explicit about that would  help. Does FC model here use importance sampling or not?

Typos
* solutions exist only few special -> only FOR few
* stochastic optional control problem -> OPTIMAL control
* drive fast than -> drive fastER
* global augment -> augmentED
* The paper would benefit from further proofreading.

---

> ### Author Response · Authors · 2020-11-18
> **Author Response to Reviewer 2**
>
> **Responses for Reviewer 2**:
>
> **1.Problematic complexity claim**
>
> We would like to thank you for pointing out the problems with time and memory complexity in the paper. The time complexity of a parallel system is sophisticated to analyze because we need certain configurations of hardware such as computation efficiency, thread synchronization time, etc. We decided to remove table 2 and only show the time and memory complexity empirically in the updated Figure 4.
>
> **2. Clarification for importance sampling**
>
> Reviewer4 raised questions about the importance sampling as well, and let me elaborate here for clarification. The main difference between our FBSDE representation and prior work (Han & Hu and the follow-up proof of convergence paper by Han, Hu& Long) is the importance sampling scheme, where the forward process of an agent is propagated using a controlled SDE where the controls are computed from explicit solutions of the Hamiltonian minimization. In Han & Hu, the forward process is an uncontrolled process. The use of importance sampling allows for more efficient sampling and hence faster convergence of the algorithm.
>
>  Since the FC architecture comes from Han&  Hu, the FC experiments do not have importance sampling implemented. We also experimented with integrating important sampling in their FC model, but it performed worse. A possible reason is listed in section 3.2 of the paper.
>
> **3. Add comparison with baseline besides qualitative results**
>
> We added a comparison against the baseline (from Han & Hu) for superlinear dynamics using the total cost, which is defined by the original objective function in eq. 2. Please refer to Figure 12 in the revised version of the paper for the comparison.
>
> **4. A possible application for numerical solver**
>
> Numerical PDE solvers based on finite element and finite difference methods do not scale beyond very few dimensions, as they rely on grids.
>
> **5. Addressed the unclear legend in the figure (5,6,7)**
>
> We updated figure 5,6,7 with a more clear legend to demonstrate the architectures we used for experiments.
>
> **6. Contributions of this work**
>
> The main advantage of the framework introduced is its scalability wrt. time and memory. The use of importance sampling and LSTM architecture accelerate training by improving the sample efficiency and introducing time correlation in the network architecture. The addition of an invariant layer renders our framework invariant to permutations of agents. This also greatly reduces the memory complexity of our framework and improves the performance when the number of agents is large. Our framework scales to 20x the problem in Han &  Hu and achieves better performance on a similar scale.

---

> > ### Comment · AnonReviewer2 · 2020-11-23
> > **Thank you for your response**
> >
> > 1. Re-framing the runtime advantage of the proposed method is definitely a plus. As a result of fixed time complexity claims I slightly increased my score.
> > 2. I would include a formal description of importance sampling in the paper.
> > 4. Would it be possible to come up with a domain that a) doesn't have an exact solution and b) is still small enough to be solved numerically? I don't know, but I am curious about that.

---

### Official Review · AnonReviewer4 · 2020-11-02
**Review on Multi-agent Deep FBSDE Representation For Large Scale Stochastic Differential Games**

**Rating:** 5
**Confidence:** 5

**Review:**



Summary:
The paper proposes a deep learning based algorithm for computing Multi agent Nash equilibrium, relying on the Forward Backward Stochastic Differential Equation representation (BSDE) of the solution together with Fictitious play algorithm. The authors present numerical examples on linear quadratic symetric stochastic games with 1000 agents, as well as solutions to partially observed 2 agents racing problem using extended Kalman filter.

Strength of the paper:
- The derivation on a scalable  Deep learning based algorithm for solving FBSDEs and related stochastic game;
- The possibility to approximate the Nash equilibrium in a game with a large number of players (more than 1000)
- The presentation of examples in a partially observable framework, where Kalman filtering is necessary to derive the solution of the game.

Weakness of the paper:
- The BSDE presented in the paper is obtained through the consideration of the corresponding Hamilton Jacobi Bellman  Partial Differential Equation, although such representation could a priori be obtained in a more general and possibly non Markovian framework
- The BSDE representation is claimed to be different form the one introduced in Han & Hu, but this should be detailed more precisely, as few information on this crucial point is given in the current version of the paper. Is it e.g. related to the use of Pontryagin principle of value process approach for the FBSDE?
- In the car racing example, it does not seem clear how the proximity to a Nash equilibrium is quantified. It is also surprising to focus on a 2 agents example, while the paper seems to intend to emphasize the scalability of the approach.
- It is a priori not clear how the presented BSDE representation differs from a related recent paper by Han, Hu and Long (Convergence of Deep Fictitious Play for Stochastic Differential Games). This should be discussed.
- It seems that this paper only focuses on games with unique Nash equilibrium, although the use of RL algorithm for games seems to become intricate whenever several Nash equilibrium shop up into the picture.


Recommandation; Given the strengths of the paper in terms of FBSDE representation of the solution, together with the weaknesses presented above, I recommend to reject the paper, but am looking forward to get clarifications on the points raised above.

Minor points:
1. Typos: « drive fast than » p8.; « invarint layer » in Figure 5;
2. Figure 2: Can you clarify the number of agents considered for such approximation? By the way, how does the solution changes as the number of agents increases? It could a priori seem to be quite stable.
3. Do the games considered here need to be non-cooperative by nature? For which purpose for the algorithm?
4. Some important references seem to be missing: some on the use of Fictitious Play for learning the Nash equilibrium in multi player games; some connecting BSDE representation to multi-player  stochastic games.

---

> ### Author Response · Authors · 2020-11-18
> **Author Response to Reviewer 4**
>
> **Responses for Reviewer 4**:
>
> **1. Issues about BSDE representation and differences between our algorithm and Han &Hu's work**
>
> While there exists a rich theory behind formulating BSDEs in a very general setting, for the stochastic differential game problem considered in this paper, the optimality condition defined by the HJB equation is the most popular and well studied. To the best of our knowledge, there is no work on non-markovian FBSDE representation for stochastic optimal control. The main difference between our FBSDE representation and prior work (Han & Hu and the follow-up proof of convergence paper by Han, Hu& Long) is the importance sampling scheme, where the forward process of an agent is propagated using a controlled SDE where the controls are computed from explicit solutions of the Hamiltonian minimization. In Han & Hu, the forward process is an uncontrolled process. The use of importance sampling allows for more efficient sampling and hence faster convergence of the algorithm.
>
> **2.Quantify Nash equlibrium**
>
> To quantify the Nash equilibrium, we have updated the results in the paper to compare the total cost of each agent (Figure 5, Figure 6). A closer proximity to the Nash equilibrium would correspond to lower cost for all agents.
>
> **3. Multiple Nash equilibrium**
>
> In this paper, we are focusing on solving problems with unique Nash equilibrium numerically and improving the scalability with respect to time and memory. Multiple Nash equilibria are not easy to solve not only for MARL but also for stochastic differential games. We believe that this is a good future direction for a potential algorithm extension.
>
> --------------------------------------------------------------------------------------------
> **Responses for minor points raised by Reviewer 4**:
>
> **1. Number of agents for invariant approximation**
>
> We consider **N-1** agents (-ith agents) for invariant approximation. The changes in the solution with respect to the number of agents can be seen in the RSE in figure 5. RSE measures the difference between the analytic solution of the value function and the approximation by the deep FBSDE framework.
>
> **2. The purpose for the algorithm and non-cooperative nature**
>
> Note that the game here needs to be non-cooperative by nature. Fictitious play is a strategy which is deployed to numerically obtain the Nash Equilibrium defined on a non-cooperative game[1].
>
> **3. Missing important references**
>
> We added several more references in the paper for learning the Nash equilibrium in multi-player games and the connection between BSDE and multi-player stochastic games.
>
> [1] Osborne, Martin J.; Rubinstein, Ariel (12 Jul 1994). A Course in Game Theory. Cambridge, MA: MIT. p. 14. ISBN 9780262150415.

---

### Author Response · Authors · 2020-11-18
**Author response to all reviewers**

**Responses for All Reviewers**:
We thank all the reviewers for critiques and suggestions.
We addressed the concerns and revised the submission with the modification marked in red.  The experiments section is updated (Figure 3, Figure 4, Figure 5, Figure 6) according to the reviewers’ suggestions and we added additional experiments to illustrate the efficiency of our algorithm (Figure 7, Figure 12, Figure 15). We address the comments of each reviewer separately.

---

### Decision · Program_Chairs · 2021-01-07
**Final Decision**

**Decision:**

Reject

**Comment:**

This paper introduces a scalable method for FSP based on FBSDE. The method is theoretically derived then applied on two problems, one simple but with many (1000) agents, and one with only 2 agents but partial observability.

The main strength of this paper lies in the scalability and the time complexity of the proposed method. Computing Nash equilibriums for many agents is a difficult problem and this paper is interesting in this aspect.

However, the reviewers point out several weak points to this paper. The difference with a previous work by Han, Hu and Long needs to be highlighted. Some parts of the paper are not clear, and too much of the important results are pushed into the appendix. Maybe this work is not best fitted to a conference format, and should be submitted to a journal? Another concern raised by the reviewers is that the experimental section does not show significant enough results, and that it is surprising to see a 2-agents problem as an illustration of a method that is aiming at addressing scalability with respect to the number of agents.

Reviewers agree on rejection for this paper, although by a small margin. I therefore recommend rejection. I think that if the authors improve this paper by following the reviewer suggestions, it can be accepted in a future venue.